# Phylogenetic Comparative Methods can Provide Important Insights into the Evolution of Toxic Weaponry

**DOI:** 10.3390/toxins10120518

**Published:** 2018-12-05

**Authors:** Kevin Arbuckle

**Affiliations:** Department of Biosciences, College of Science, Swansea University, Swansea SA2 8PP, UK; Kevin.Arbuckle@swansea.ac.uk; Tel.: +44-179-260-2087

**Keywords:** venom, poison, evolution, ecological toxinology, comparative biology, data analysis, methodology, phylogeny

## Abstract

The literature on chemical weaponry of organisms is vast and provides a rich understanding of the composition and mechanisms of the toxins and other components involved. However, an ecological or evolutionary perspective has often been lacking and is largely limited to (1) molecular evolutionary studies of particular toxins (lacking an ecological view); (2) comparisons across different species that ignore phylogenetic relatedness (lacking an evolutionary view); or (3) descriptive studies of venom composition and toxicology that contain post hoc and untested ecological or evolutionary interpretations (a common event but essentially uninformative speculation). Conveniently, comparative biologists have prolifically been developing and using a wide range of phylogenetic comparative methods that allow us to explicitly address many ecological and evolutionary questions relating to venoms and poisons. Nevertheless, these analytical tools and approaches are rarely used and poorly known by biological toxinologists and toxicologists. In this review I aim to (1) introduce phylogenetic comparative methods to the latter audience; (2) highlight the range of questions that can be addressed using them; and (3) encourage biological toxinologists and toxicologists to either seek out adequate training in comparative biology or seek collaboration with comparative biologists to reap the fruits of a powerful interdisciplinary approach to the field.

## 1. Introduction

Toxinologists have long focused on animal weaponry such as venoms and poisons, and much has been learned about the composition and mechanisms of substances from a wide range of species [1,2,3,4,5,6]. However, an explicit focus on the evolutionary and ecological aspects of venoms and poisons has been a relatively recent development [7,8] despite the promise of such an approach to explain the compositional and functional patterns evident in the literature. That is not to say that the subject has been entirely ignored previously, but that it has rarely been a mainstream area of toxinology. Indeed, even much of the recent work in evolutionary and ecological toxinology has involved descriptive work on venom composition and/or function with a post hoc ecological or evolutionary interpretation added to discussions. For instance, some aspect of the venom of a group of snakes could be described and the results may show that one species is an outlier in some way. The researcher may be tempted to look for unique attributes of that species, and noting that it is the only bird-eater in the group may interpret the venom differences in terms of adaptations to feeding on birds. The descriptive nature of many such studies certainly has a place, but as Platt recognised in his classic paper on strong inference in science [9] we need to incorporate studies framed solidly as tests of hypotheses if we are to truly advance our understanding. Since a great deal of variation necessary for testing evolutionary or ecological hypotheses concerning venom evolution occurs across different species, in the current paper I aim to introduce toxinologists to comparative biology (the study of evolution using interspecific data) as one underused approach to strengthen our understanding of the evolutionary ecology of venom and poison.

Is there really a lack of comparative biology in evolutionary toxinology? After all, this approach is characterised by the use of phylogenies and both phylogenetic trees and the word ‘comparative’ are frequent occurrences in evolutionary toxinology papers. Unfortunately, the word ‘comparative’ is often used to refer to studies involving more than one species, which is indeed comparative biology in the broad sense, but often without taking an appropriate approach to analyse these data and making inferences based on essentially a comparison of two species (*N* = 1 comparison). Furthermore, many of the phylogenies used are trees of particular toxins, which are correctly used to consider the molecular evolution of that particular class of molecules but are not the species phylogenies necessary to gain an ecological framework nor do they necessarily relate to the evolutionary history of the venomous species (and so lack consideration of the evolution of the venom system as a whole). Occasionally, species phylogenies are included in papers but are used only visually to illustrate the ideas of the paper and are not integrated into the analysis by taking a truly comparative approach. The comparative approach is essential for analysing interspecific data (as I’ll explain below), and as Joe Felsenstein succinctly put it in his classic paper formalising the comparative method in 1985 [10]: “Phylogenies are fundamental to comparative biology; there is no doing it without taking them into account”. Note that data from, and intraspecific phylogenies of, different populations are equally amenable to analysis with comparative methods. Hence, the focus on the more common case of interspecific data in this article should not be taken as indicating that intraspecific phylogenetic data are unimportant as the same comments apply where there is important interpopulation structure.

With the above considerations in mind, the current situation using a survey of articles published in *Toxins* will be illustrated. Only research articles (i.e., excluding other article types such as reviews, commentaries, etc.) published in the ‘Animal Venoms’ section of the journal will be considered as this section is most relevant to the evolutionary and ecological questions for which the comparative approach is insightful. All of these between the first published in 2012 and the date on which the survey was conducted (27 October 2018) were included, with the exception of a very small number of articles which covered only microbial toxins (less relevant and probably not best suited to the ‘Animal Venoms’ section in the first place). This survey recorded whether the paper focused on multiple species rather than on a single species which is simply compared to others used only as references or which focus on the molecular evolution of particular toxins (e.g., 3FTX) and, therefore, not on the venom of a particular set of species. These ‘multispecies’ papers are those which could reasonably benefit from a comparative approach. It should be noted that (1) the questions addressed by some of these papers may not necessarily benefit from a comparative approach, and (2) where the focus is on molecular evolution of toxins (including those with toxin trees), there are still cases where inferences are made about where/when toxin types originated on the tree, and these are comparative questions. Hence, the summary presented here is not intended to critique any of the papers involved but rather to gain an insight into the broad state of the field. Papers focusing on multiple species recorded whether they presented only trees of toxins (not species as well), whether they presented species phylogenies, and whether they used a formal comparative approach to their analyses. Also, the number of species included in multispecies papers, which represents the sample size for comparative inferences, was recorded. We would expect that, with a few exceptions, the number of comparative papers should approximate the number of papers presenting phylogenies. This is because comparative biology requires phylogenies and if a phylogeny is available and presented it contains information that should inform analyses using a comparative approach. Moreover, we would hope that the majority of multispecies papers have used a comparative approach as this is the only reasonable way to draw strong inferences from interspecific data [10] and that this is done with a reasonable sample size (number of species)—these data reflect how well multispecies studies are designed to address comparative questions.

The results from this survey of papers published in *Toxins* shows some interesting patterns (Figure 1).

Firstly, despite the journal publishing papers in a wide range of areas that would generally be focussed on one species (e.g., treatment of envenomation, molecular function, potential applications of toxins, molecular evolution of toxin classes), 18% of papers were focused on multiple species. This reflects the increasing attention given to the evolution and ecology of venoms and consequent comparisons of multiple species and highlights the potentially wide reach of comparative methods. Despite this, 70% of these did not use phylogenies in any way (magnified to 76% when the author’s own publications are excluded) and only 5 papers (14% of multispecies papers) made use of comparative methods to test questions on their interspecific datasets (dropping to only 2 papers, 6%, when mine are excluded). This suggests that the necessity and utility of the comparative approach when analysing data from multiple species to understand the evolution and ecology of venoms is remarkably underused. A notable concern here is that most multispecies studies contained far too few species (independent replicates) to adequately test any questions based on species comparisons. In fact, a quarter of multispecies studies only contain 2 species despite most of them suggesting evolutionary or ecological explanations for the differences found that are unsupported by *N* = 1 comparisons [12].

The importance of including phylogeny in the analysis of interspecific data has been described well many times before [10,13], but I will give a brief illustration of why this is necessary. Importantly, accounting for phylogeny does not inevitably lead to weaker relationships between the variables of interest, but can also lead to stronger ones (Figure 2). This is because the evolutionary relationships between the taxa in question contain information that can make the statistical relationships between the variables in an analysis either more or less ‘surprising’ than they seem without accounting for phylogeny.

When testing any hypothesis using statistical analyses, there is a basic assumption that the data points are independent of one another in the sense that knowing the value of one data point does not give you information on which value another should take. As a result, each data point contributes the same amount of information towards testing a hypothesis when compared with others. However, if the observations (data points) are taken from different species which are related to each other in varying degrees due to their phylogenetic history, then we expect a priori that more closely related species will be more similar in the traits we are measuring, and so they are not independent. We need methods to account for this dependence and give us results which are robust to the patterns we expect simply as a consequence of evolutionary relatedness (with no further relationship between the variables of interest). This is the essence of the traditional ‘comparative method’ (see Section 2 for more information). However, comparative biology (and hence the suite of comparative methods available) aims to do more than simply account for the influence of phylogeny in our standard statistical analyses. Phylogenies contain information on the timings and patterns of lineage splitting in our group of interest, and as such essentially they represent the historical vehicle that has carried the group’s traits as well as their populations to the present day. We can use this information in combination with whatever traits we are measuring to ask a much broader range of questions about the evolution of the system, discussed in more detail in Section 3, Section 4, Section 5, Section 6 and Section 7 below. Hence, if toxinologists at least become familiar with what questions are possible it may lead to the development of the field towards a greater understanding of the evolution and ecology of venoms and poisons.

This review offers a brief introduction to the types of comparative approaches available and uses examples from studies on venoms and poisons to show how these different approaches can be applied by toxinologists. It makes no attempt to provide a comprehensive introduction to comparative biology as this would require an entire book-length treatment, but provides references to introductory reviews for each section of this review in Table 1. This table also provides a non-exhaustive list of packages available in the R statistical environment [14] for conducting these analyses, the focus on R being a result of the rapid development of methods and associated packages in that software. For a more extensive curated list of R packages see the CRAN Task View for Phylogenetics (available at https://cran.cnr.berkeley.edu/web/views/Phylogenetics.html). Nevertheless, comparative biology is a specialism in itself and toxinologists are encouraged to either receive some degree of training in comparative biology or consult a comparative biologist before running such analyses (and preferably at the earlier stage of planning the study).

## 2. Accounting for Phylogeny in Statistical Analysis

By far the most common use of phylogenies in statistical analysis is to account for the effect of phylogeny, so much so that this class of methods is often simply called ‘the’ comparative method. In Section 1 it was shown why accounting for phylogeny in data analysis is important, and fortunately there are phylogenetic comparative ‘equivalents’ for many standard statistical tests. In the same way as non-phylogenetic statistical tests, these can be used to test the relationships between two (or more) variables that we might be interested in, but we can also account for the effect of phylogeny.

Many hypotheses for which standard statistical tests are used can be formatted as general(ised) linear models (GLMs), including t-tests, regressions, n-way analyses of variance (ANOVAs), logistic regressions, and Poisson regressions. GLMs essentially all take the form *y* ~ *x*_1_ + *x*_2_ + … + *x*_n_ whereby a response variable (*y*) is predicted by one or more explanatory variables (*x*’s). When the response variable is continuous the number and quality (categorical or continuous) of the explanatory variables vary to create what are effectively *t*-tests (one categorical), regressions (one continuous), or 2-way ANOVAs (2 categorical), but the general model structure makes clear the relation and expandability of GLMs. Similarly, applying transformations to the response variable within the fitting of the model allows non-continuous response variables such as are used for logistic or Poisson regressions (all following the same generalisable framework of the GLM).

GLMs assume that the residuals of the model are uncorrelated (independence of data points), but if we have an expected correlation structure we can build this into the model. For interspecific data, that expected correlation structure is provided by phylogenies—these contain a representation of how much evolutionary history each pair of species share compared to others. Consequently, phylogenetic equivalents of GLMs exist in the form of pGLS (phylogenetic generalised least squares) [46,47], phylogenetic logistic regressions [48], and phylogenetic Poisson regressions [49]. Technically the latter are not GLMs but rather GEEs (generalised estimating equations); however, for the purposes of choosing appropriate phylogenetic equivalents we can consider phylogenetic Poisson regressions as similar to GLMs. These models, therefore, represent a very general framework for analysing interspecific data that is intuitive to interpret for those already familiar with standard statistical tests.

Some readers will also be familiar with generalised linear mixed-effect models (GLMMs) which are commonly used to deal with data sets containing repeated measures from individuals by modelling variables (in that case individual ID) as ‘random effects’ which build structure into the residual variance of the models. Phylogenetic GLMMs (pGLMMs) also exist which include the phylogeny as a random effect and so estimate the proportion of variance in the response variable attributable to evolutionary history and in doing so account for this in the model [50]. In part due to the highly flexible implementation of pGLMMs in the MCMCglmm R package [26] this method can be very useful as it can incorporate a wide range of distributions of the response variable including ordinal traits, can have multiple response variables (generating multivariate pGLMMs), and can handle a wide range of other extensions.

A range of other methods exist which fall into the same group of ‘phylogenetic equivalents to standard statistical tests’, including phylogenetic paired and one-sample *t*-tests [51], phylogenetic principal components analysis [52], and a wide range of phylogenetic comparative methods for interspecific geometric morphometric datasets [53]. The explosion of comparative methods prevents comprehensive coverage of those available, but it is worth being aware of the fact that few commonly used statistical tests lack a phylogenetic equivalent that can be used for analysing interspecific datasets.

Because this group of comparative methods is so broadly applicable to any question which you would use other statistical tests on for intraspecific data, these provide the most useful tools for toxinologists to be aware of. Indeed, they have already been used many times to test hypotheses regarding animal venoms. For instance, pGLS has been used in several clades of Australian elapid snakes to investigate relationships between venom toxicity and diet [54,55], the cofactor dependence of toxic effects [54], the relationship between venom components and toxic effects [56], and the relationship between venom toxicology and antivenom efficacy [54]. As discussed in Section 1, drawing inferences about statistical relationships between traits such as coagulotoxicity and diet [54,55] is prone to being misleading when evolutionary history is ignored. Hence, phylogenetic comparative methods provide a powerful way to strengthen our understanding of the ecology, toxinology, and medical treatment of envenomations.

## 3. Ancestral States

The estimation of ancestral states remains a frequent goal of evolutionary studies, including those in toxinology, and it is also among the most widely known forms of comparative biology. However, in many toxinological studies which aim to investigate ancestral states, say the ancestral venom composition of a group, there is no attempt to formally estimate them. Instead, descriptive studies of venoms from different species are presented alongside a phylogeny. The latter frequently has a ‘mapping’ of traits (e.g., toxins) onto the tree, but this is often done simply by eyeballing the phylogenetic distribution of the trait. This approach typically ends up resembling an estimate using maximum parsimony to varying degrees, as this is the most ‘intuitive’ and simple method of estimating ancestral states which seeks to minimise the changes over the tree. However, this is a poor strategy since errors are easy to make without formal inference and maximum parsimony (although intuitive) can often perform poorly compared to likelihood-based estimation [28,29], which often arrive at an estimate involving more change than parsimony would suggest. There are many methods for formally estimating ancestral states and toxinologists should make use of them rather than relying on mere visually-informed assumption.

Likelihood-based methods for estimating ancestral states require a model (as for all likelihood-based statistical methods), and the need to estimate an appropriate model to base the estimate on is one limitation of such methods. Nevertheless, in the next section I will introduce a range of trait evolution models that can be evaluated against each other for a given dataset to decide on the best fit. This empirically chosen model with its estimated parameters can then provide an informed framework for estimating ancestral states.

In some cases the ancestral states may be of interest in themselves, but they also provide the opportunity to test the temporal order of evolution either between different states of the same trait or across different traits. This consequence of estimating when and where on the phylogeny a given set of changes happened can be of direct interest or can provide powerful supplementary information towards inferring the direction of causality of a relationship between traits. If two traits are associated with each other across a phylogeny and you have reason to expect this association to be causal, then if one consistently appears earlier in the phylogeny than the other it provides added support for that one causing the change in the other. Note, however, that ancestral state estimates are not direct tests of causality and cannot stand on their own as strong evidence of a causal (or even non-causal) link.

Associated with the poorness of the ‘eyeballing’ strategy noted above, maximum parsimony performs especially poorly when traits evolve quickly. This is because it assumes that the minimum change possible to produce the pattern seen on the phylogeny, but traits which evolve rapidly will change state multiple times across their evolutionary history—a scenario invisible to parsimony but estimated by likelihood-based approaches. This is particularly important for animal venoms since many of these show rapid rates of evolution [11,57,58]. A related benefit of likelihood-based estimates of ancestral states is that, unlike parsimony, they provide a means of quantifying the uncertainty surrounding the estimates. The magnitude of such uncertainty can be great, particularly for continuous traits which are more difficult to estimate accurately than categorical traits due to the greater number of possible states (essentially infinite for continuous traits, though with varying plausibility, and constrained to the number of categories assigned to discrete traits). It is especially important for quickly evolving traits, since this increases the uncertainty further, to acknowledge the level of uncertainty in our estimates and present this alongside the estimates where possible. Explicitly acknowledging the uncertainty associated with our estimates is standard practice in any area of data analysis and ensures our inferences are well-grounded and appropriately interpreted.

One example of the use of ancestral state estimation to investigate the evolution of venom systems illustrates this well [59]. In that study, morphological traits of the venom system of 34 hymenopteran insect species, such as the degree of serrations on the stinger, were investigated using techniques including ancestral state reconstruction. The best (point) estimates of the traits were plotted across the tree and the 95% confidence intervals were added to all nodes. Although sufficient estimates were made to draw inferences from these reconstructions (for instance the locations and minimum numbers of origins of strongly serrated stingers), except for traits showing little change across the phylogeny it is notable that confidence intervals widen very quickly as the relevant nodes get further away from the tips. This is a common finding for ancestral state estimations of toxicological traits of venoms, perhaps even more so than morphological attributes of venom systems. Nevertheless, when appropriate care is taken in interpretation, ancestral state estimation can give substantial new insights into the evolution of venom systems.

## 4. How Does It Evolve? Trait Evolution Models

Using a phylogeny and data on traits (for instance presence of particular toxins, effectiveness of a given antivenom, cytotoxicity, etc.) we can construct and compare different models of trait evolution to better understand the dynamics of the trait over time. This can give us an insight into the probability of different evolutionary outcomes (e.g., are some changes very rare, possibly very difficult, compared to others?) and provide a better understanding of the evolutionary processes and patterns involved. There is a wide range of different models that can be used, but the first consideration is whether the trait you are interested in is continuous or categorical. This paper will briefly highlight some particularly important or interesting models below as an indication of what type of hypotheses we can test.

Continuous traits are essentially modelled as constantly changing over evolutionary time to some degree. This is expected since continuous traits are likely to experience at least minor evolutionary change due to drift and the combined architecture of (usually) being controlled by many genetic loci and often a range of environmental factors. In other words, continuous traits are unlikely to remain constant for any period of evolutionary time.

The simplest and most frequently used model for continuous trait evolution is Brownian motion (BM) [10,60], which consists of a single estimated parameter (σ^2^) representing the variance per unit time, in other words the rate of evolution. BM is often used as a form of ‘null model’ (though its application as a true null model depends on the question being asked) and predicts that evolutionary change can either lead to an increase or a decrease in the trait value at each time step, with equal probability. This is frequently interpreted as ‘random evolutionary change’ or support for genetic drift, but care must be taken with such inferences since other processes are also consistent with BM patterns (e.g., selection towards a randomly varying optimum trait value).

The Ornstein–Uhlenbeck (OU) model [47,61] can be thought of as an extension of the BM model and also estimates a parameter for the evolutionary rate of the trait. However, whereas variance continues to accrue through time and the trait value is equally likely to increase or decrease under BM, OU incorporates a parameter (α) representing the strength of a ‘pull’ towards an optimal value (θ). Under an OU model, the further the trait value moves away from the optimum the stronger the pull back towards it—somewhat analogous to an elastic band connected the fixed optimum value to the varying trait value. For a given deviation from the optimum value, a larger α value will make the trait more likely to change in the direction of the optimum. The evolution of the trait is, therefore, constrained around a particular value in the OU model, and has consequently been used to represent evolution under selection. However, as with BM care must be taken in such a process-based interpretation since selection only represents one type of constraint that could lead to evolution around a particular trait value. Since the α parameter represents the strength of the ‘pull’ towards the optimum, it should be clear that when this parameter is 0 the OU model becomes identical to the BM model (and tends towards BM when α is low).

Although the BM and OU models are by far the most commonly used to study trait evolution, another model may be of particular interest to evolutionary toxinologists—the early burst (EB) or its more general form the ACDC (accelerating/decelerating) model [62,63]. This model allows the rate of trait evolution to vary through time, either starting slow and getting faster (accelerating), or starting fast and slowing over time (decelerating or ‘early burst’). In particular, the EB version of the model reflects the expected pattern under adaptive radiation or other processes leading to rapid exploitation of newly occupied niche space. Evolution will favour trait diversification until niches are filled and then (as selection for this diversification is relaxed) a gradually decreasing rate of evolution. Since chemical weaponry is typically strongly related to exploitation of food resources and/or should relax behavioural constraints imposed by predation pressure [64,65,66] we might expect the enhanced access to ecological opportunities to lead to venom and poison traits evolving under EB models. Indeed, a pattern in which older venomous lineages (whose toxins been evolving for longer) show slower rates of toxin evolution than younger venomous lineages has been recovered in a broad-scale analysis [58], although without explicit modelling of a slowdown in trait evolution. Nevertheless, a recent study on poison dart frogs (*Ranitomeya imitator*) found no evidence for an ecological release from predation pressure in more highly toxic frogs, at least in the context of calling site selection [67], so broad scale comparative studies are needed to evaluate the importance of toxic weaponry for ecological opportunity.

Categorical traits (e.g., presence vs absence of venom or particular toxins) are typically modelled as a Markov process (sometimes known as Mk models) in which the trait has a probability of changing from one state to another during each time unit [13,33,34]. These probabilities are estimated as transition rates for each possible transition between states, and so can allow for distinctly different models to be tested. The most general version is sometimes called the ‘all rates different’ (ARD) model and simply represents a model where all transition rates are allowed to take different values, such that the probability of evolving from state A and state B is not necessarily the same as the probability of moving from state A to state C, or from state B to state A. Other Mk models are special cases of the ARD model with various constraints imposed. These models are the basis of many evolutionary pathway models for categorical traits, but as these are covered in the next section only two other Mk models will be introduced here. The other extreme from the ARD model is often called the ‘equal rates’ (ER) model and consists of a single transition rate for all possible transitions—essentially assuming that changes from one given state to any other state are equally likely. Between these two extremes lies the ‘symmetrical’ (SYM) model, in which forward and backward transitions have equal rates for each pair of states. For instance, the probability of changing from state A to state B is the same as the probability of changing from state B to state A, but can be different from the probability of changing from state A to state C. Note that for categorical traits with only two states (e.g., presence vs absence), the ER and SYM models are identical, but they differ when traits have more than two states.

Another model for categorical trait evolution which has been recently described is the threshold model [68,69]. This differs importantly from the Mk models above in being a model with ‘memory’—The probability of changing state is dependent on how much time has passed since the trait last changed state. Put simply, just after a trait evolves to a different state it is more likely to evolve back to the original one, and becomes less likely over time. This may be plausible for many categorical traits if they are underlain by many genetic loci (either coding directly for the trait or involved in pleiotropic or epistatic interactions) or are one trait amongst a coordinated suite of traits, of which other members may not have undergone evolutionary change yet. The threshold model is implemented as an (unobserved) continuous trait called ‘liability’ which evolves according to a BM model, and which causes transitions between the observed categorical states when it crosses a certain threshold. Because just after crossing a threshold (changing the state of the categorical trait) a BM model is as likely to go back across it as further away from it in the next time unit, this creates the property of ‘memory’. The threshold model has only recently been widely recognised in comparative biology and so has not yet been extensively used, but can clearly be applied to many cases of trait evolution including those in toxinology.

Beyond simply comparing the fit of different models (whether continuous or categorical) to understand the evolutionary dynamics of traits, many hypotheses concern distinct shifts in the evolution of a trait. These may be shifts in the type of model involved, perhaps we predict that a certain type of toxicity (say neurotoxicity) should generally evolve by BM in the context of other toxin types, but that during an evolutionary event that causes neurotoxicity to be strongly favoured (perhaps after specialising on a new prey type) the extra selection pressure should shift the toxicity to evolving under an OU model. Alternatively, we may predict shifts in the parameters of one type of model at some point in the phylogeny, perhaps expecting a much higher rate of loss of venom in a lineage which switches to specialising on a diet that does not need to be subdued (e.g., the classic toxinological example of egg-eating seasnakes [70]). Comparative methods exist to detect such shifts in evolutionary models and could be effectively used to address a wide range of questions in evolutionary toxinology [22,30,45]. However, even studies that have considered trait evolution in animal venoms and poisons have only estimated and interpreted parameters of general versions of the models [11]. There remains great potential for exploring the fit of alternative models to better understand the evolution of toxic weaponry, but this remains amongst the most underused group of comparative methods in toxinology.

## 5. How Did It Get to What We See Today? Evolutionary Pathways

Following on from the discussion of Mk models in the previous section, the ability to construct such models with any given constraints allows us to specify alternative evolutionary pathways and compare the fit of models representing each one [33,34]. Essentially, we can force the model to estimate transition rates between states of a categorical trait for a given pathway by constraining some rates to equal 0. For instance, if we have three states (A–C) and we want to test how a species goes from having state A to state C we can construct models representing the three possibilities. We could hypothesise (1) an ARD model in which both A->B->C and A->C are possible, (2) a model with the transition rate from A to B equal to 0, such that we can only go directly to C (A->C), or (3) a model with the transition rate from A to C equal to 0 such that the only route between the two is via an intermediate stage in state B (A->B->C). Comparing the fit of these three models to the trait data and phylogeny can allow us to answer our question.

Pathways can also be coded into structural equation models in the form of phylogenetic confirmatory path analysis [71,72]. This is a method that attempts to establish causal relationships from a series of GLM-style analyses (see Section 2) and so can evaluate the causal relationships between several different traits, which in themselves can take different forms (e.g., binary, continuous, etc.). Phylogenetic confirmatory path analysis, therefore, represents an excellent way of testing evolutionary relationships between different, potentially causally linked, traits and so could be considered a method to address the evolutionary pathways of traits.

In the course of a comparative study on the conservation status of poisonous amphibians [73], a relationship was found in which chemically-defended amphibians are more likely to be threatened than others. Since it is difficult to identify the direction of this relationship from the GLM-style analyses (see Section 2) which support the link, a one pathway model was created in which a non-poisonous and non-threatened amphibian lineage could evolve to be a chemically defended and threatened one by changing either trait first, and another model that constrained chemical defence to evolve first (so defence changes before threat status does). Comparing the fit of these two models provided strong evidence that toxic weaponry can negatively impact the extinction risk of amphibians. This type of directional inference cannot be provided with non-phylogenetic approaches which lack temporal information and so highlight the type of questions that can be addressed by these methods.

## 6. Convergent Evolution

The independent evolution of similarity, or convergent evolution, is well known as a common feature in the evolution of animal venoms [2,7], and the resistance of some animals to toxic defences has become a textbook example of convergence [74,75]. Convergent evolution is an important feature in itself as it relates to our understanding of the predictability of evolution, but importantly for the purposes of this review it also presents an opportunity to understand how venom systems evolve in much greater detail. For instance, studying patterns of convergence in cobras and related elapid snakes provided evidence not only for multiple independent origins of cytotoxic venoms and associated defensive characteristics, but also that cytotoxicity was more strongly convergent in hooding species than in spitting species [76]. Quantifying convergent evolution is a crucial step to testing many hypotheses about the evolutionary drivers and patterns of convergence [38], but requires a phylogenetic comparative approach to enable meaningful quantification. Since ancestral state estimations showed that hooding in cobras evolved before cytotoxicity, the comparison of evolutionary convergence in toxicity between hooding and spitting cobras provided evidence that the defensive display behaviour of hooding has been an important driver of cytotoxicity in cobra venoms (perhaps even more so than spitting behaviour) [76].

Convergent evolution of venom systems has typically followed the ‘mapping by eyeballing’ strategy discussed in Section 3, but formal ancestral state reconstruction can provide a way to formally identify cases of convergence (see Section 3). Moreover, methods specifically designed to investigate convergent evolution have been developed [37]—this paper will very briefly highlight some of these here but see Arbuckle et al. [37] for a more detailed summary. These methods aim to either identify or quantify convergence, the latter being a key aim for a more detailed understanding of the system in question [38].

One approach specifically designed to identify cases of convergence is SURFACE [77], which makes use of the OU models discussed in Section 4. SURFACE takes a phylogeny and data for traits (although it will run on a single trait, the performance of the method is improved with more traits per analysis) and fits an OU model across the tree. It then attempts to fit a second OU model (with different parameters) to each part of the tree, creating a ‘regime shift’, and continues adding regimes until no more are statistically supported (using Akaike’s information criterion). Finally, SURFACE attempts to combine as many of these different models into groups of models with the same parameters, such that the same ‘regime’ (fitted OU model) can occur multiple times throughout the phylogeny (i.e., to allow convergent evolution). Convergence is then identified as statistically supported shifts to the same OU model in different parts of the phylogeny. SURFACE also allows the quantification of the frequency of convergence via the number of convergent shifts, the number of different convergent regimes, and the proportion of regimes that are convergent.

A more graphical method of identifying convergence is the ‘phylomorphospace’ plot [36]. This is simply a standard (scatter) plot of two or more continuous traits from different species onto which a phylogenetic tree is projected connected the data points for each species. Estimating ancestral states and plotting these onto the phylomorphospace allows tracing the evolutionary history of the traits across the plot. Identification of convergence is then simply carried out based on multiple branches independently appearing in the same area of the plot. A set of ‘C-metrics’ have been designed to quantify the frequency and the ‘strength’ (magnitude) of convergent evolution based on the movement of the phylogeny over the phylomorphospace plots [36].

The Wheatsheaf index provides a method for quantification (but not identification) of convergent evolution, specifically the strength of convergence [78]. This metric differs from others in that it measures convergence in one or more traits in relation to a binary trait possessed by, and defining, the ‘focal group’. This allows comparison of the strength of convergence in a set of traits for a particular niche or a particular attribute – as was used to quantify convergence of cobra venom in relation to first hooding and then spitting in the earlier example in this section [76]. The Wheatsheaf index combines two aspects of trait evolution. Firstly, it considers convergence to be stronger when members of the focal group are more similar to each other. Secondly, convergence is considered to be stronger when the focal group are more dissimilar to the non-focal group, as this implies a stronger selective pull across the adaptive landscape.

An important point to note in quantitative studies of convergent evolution is that the traits for which we want to measure convergence are always continuous. This is because measuring convergence in categorical traits (especially binary traits) remains a major unsolved, and perhaps unsolvable, challenge [38]. Beyond measuring the frequency of convergence it is difficult to envisage how a meaningful quantification could be achieved for binary traits—they are either the same or not. Nevertheless, many of the traits we are interested in as evolutionary toxinologists are continuous—toxicity, yield, and diversity of venoms and poisons, neutralisation ability of antivenoms, etc.—and so there is great scope to incorporate comparative methods designed for these questions into our studies.

## 7. How Does It Relate to Evolution of Lineages? Diversification Dynamics

The interplay between the speciation and extinction rates of a lineage of organisms define the (net) diversification rate [75]. Specifically, net diversification rate is the speciation rate minus the extinction rate. These rates are not necessarily uniform across different lineages or through time and this variation is referred to as the diversification dynamics of a lineage. Diversification clearly has fundamental importance for evolutionary biology as it results in the diversity of life we see today, in terms of both the diversity of species (species richness) and the diversity of traits (and their distribution over a phylogeny). An important point is that diversification rates are not always intuitive based on species richness, since species richness is an interaction between diversification and time. For instance, marine fish have much higher species richness in the tropics but have much higher diversification rates in the polar regions [79], which have been colonised more recently and so have not had time to accrue species. Hence, formal analysis of diversification dynamics is crucial to understand the observed patterns.

Diversification dynamics can be shaped by the traits possessed by lineages [44]. For instance, ‘key innovations’ are traits that lead to faster diversification rates [80] either by increasing speciation rates or by decreasing extinction rates, perhaps by increasing ecological opportunity, dispersal ability, or other important factors in diversification. Alternatively, traits may have the opposite effect and increase extinction rates (or decrease speciation rates) leading to lower diversification rates in lineages that possess them. Clearly, since diversification involved macroevolutionary events such as speciation and extinction and occurs over evolutionary time, questions relating to diversification dynamics rely on phylogenetic comparative methods and are inaccessible to other approaches (excepting fossil data in some cases).

So is there reason to think that toxic weaponry should influence diversification? Yes—there are many interesting questions here that evolutionary toxinologists can answer using comparative methods. In particular, the defensive roles of poisons and venoms are expected to lead to increases in diversification rates, since ecological opportunity is thought to be often constrained by predation pressure that effective defences can overcome [81]. The majority of evidence for this pattern comes indirectly via higher diversification rates in aposematic species (those which advertise an antipredator defence with warning signals) [82], rather than an effect of toxic weaponry itself. In fact, the potential influence of aposematic signals is such that even non-toxic species which mimic venomous or poisonous species may increase their diversification rates simply by possessing the signal—one of the proposed ‘Savage–Wallace effects’ [83]. These far-reaching effects are predicted to result from mimicry enabling harmless species to occupy more ‘dangerous’ (higher predation risk) niches and lifestyles which may enhance diversification. Earlier evidence for higher diversification rates in plants with latex production as a chemical (though non-toxic) defence did not withstand reanalysis [84], but recent work on venomous and poisonous animals suggests increased diversification rates in tetrapods that use toxic weaponry, with the exception of amphibians [11]. In contrast, poisonous amphibians had lower diversification rates as a result of higher extinction rates than non-chemically-defended species [85]. The reasons for the increased extinction rate of poisonous amphibians over evolutionary time is still unknown, but persists into the present day as increased probability of having a threatened conservation status [73]. These results open up a new avenue for research to understand not just why poisonous amphibians suffer higher extinction risk, but why this differs from other toxic tetrapod groups. This result would not be clear nor would its further exploration be possible without an explicitly phylogenetic approach. There are still many questions waiting to be answered concerning the relationship between toxic weaponry and diversification dynamics, providing fertile ground for comparative toxinology.

Many comparative methods exist for investigating diversification dynamics, and these can essentially be considered as forming two groups: those which analyse diversification without reference to traits (trait-independent), and those which explicitly test for associations between a trait and diversification (trait-dependent). For questions related to toxic weaponry we would typically be interested in relating traits to diversification, however even trait-independent methods can be useful. For instance, using methods that can look for shifts in diversification across a phylogeny [86,87,88,89] we can identify where these shifts occurred and compare those locations to trait origins estimated with ancestral state methods (see Section 3). Alternatively, trait-dependent diversification methods are widely available and can either take the form of estimating diversification rates to compare with the evolution of the trait [31,90,91,92] of making sister group comparisons [93]. There are a few variations of sister group tests but they all essentially work on the principle that sister-groups are the same age by definition, and so differences in species richness between them reflect differences in diversification rate. Based on this, sister group methods find many pairs of sister groups which differ in the trait of interest and test whether the groups with the trait consistently have more or fewer species than their sister groups without it. Comparative methods for diversification are amongst the most controversial and debated areas of comparative biology so the author strongly recommends discussing the planned approach with a comparative biologist before choosing a method for a particular study.

## 8. Potential for New Tailored Methods

Toxinologists are not limited by previously developed methods—the impressive expansion of phylogenetic comparative methods in recent years is testament to the diverse range of questions that can be addressed in a range of different fields. If an appropriate approach is not available then work on developing new comparative methods to solve problems is encouraged. These may entail entirely new models or simply a new workflow of existing methods to open up avenues of research.

As way of explanation of the ‘new workflow’ approach this paper will provide a suggestion that could be pursued in more detail in future work. Perhaps we want to quantify the similarity of whole venom profiles across a group of species and investigate whether the profiles are predominantly an effect of phylogeny or more strongly influenced by something else, perhaps diet. One way of evaluating the overall diversity of a given venom would be to conduct 2D-PAGE and observing the pattern on the gel. In the field of animal coloration, methods have been developed to quantify the overall similarity of patterns (e.g., the distance transform method [94]), and these could likely be applied equally well to the patterns on a gel. Of course, this would require attempts to ensure the gels are appropriately aligned such that a mark of the same size in the same location in two different images represents meaningful similarity. This requirement may be variably feasible but if achieved then pairwise quantitative distances between overall venom profiles would be obtained. Standard statistical methods for cluster analysis could be used to obtain dendrograms representing ‘phenotypic trees’ that can be compared with phylogenetic trees of species using ‘tree distances’ [95,96] or methods developed for coevolutionary or ‘tanglegram-based’ convergence studies [97,98,99,100]. Although not well developed here, this suggestion highlights how new comparative methods do not necessarily need developed from scratch and that they can be developed to target problems specific to toxinology (or any other field).

## 9. Visualising Results

When using standard statistics, it is widely acknowledged that plotting results in some way is an important part of the data analysis. Visualising the data in a way that shows the results does not only provide an intuitive way to understand what the statistical results are telling us, but also allows us to ensure that the data look appropriate. For instance, plotting can reveal problems in our assumptions or influential outliers, as in the classic ‘Anscombe’s quartet’ dataset [101]. Recently, Liam Revell and colleagues have produced a similar (but comparative) dataset that serves to emphasise that comparative methods are no exception [102]. In fact, because interspecific datasets are necessarily complex due to the structured nature of phylogenies, visualisation of results is even more important to allow clear inference.

Fortunately, a wide range of methods for visualisation results from phylogenetic comparative analysis are available [103,104] (see Figure 3 for examples). Many of these also produce attractive figures which is a useful trait for science communication and engagement in addition to their benefits for scientific interpretation. For GLM-style analyses (see Section 2) it is possible to plot the raw data and a regression line from the analysis which accounts for phylogeny. However, this approach has a cost of intuitive interpretation since the line may not seem to fit the data well when the reader cannot consider the phylogeny simultaneously. Consequently, it is likely preferable in many cases to plot the tree and the data together, particularly when only one trait is considered in the analysis [102]. Some common options for this include plotting estimates of ancestral states over the tree (for continuous traits), plotting estimated probabilities of a given state over the tree (for categorical traits), plotting pie charts at nodes of the phylogeny to display the probabilities of each (categorical) state at the nodes, plotting ‘traitgrams’ and phylomorphospace plots (see Section 6) of continuous traits, and plotting data next to tips of trees. This far from an exhaustive list illustrates the diversity of approaches available, and multiple methods can be combined in a single figure. For instance, Panagides et al. [76] displayed ancestral state estimates for three separate traits relating to cobra venom and defensive behaviours on single figures; using branches coloured to represent a continuous trait and two categorical traits shown as pie charts at nodes (one above the branch and one below the branch).

Although many visualisation methods for comparative biology do not yet readily incorporate the uncertainty around the estimates, where this is possible it should be shown. Point estimates can give a false sense of confidence in results which leads to an overly strong interpretation, and incorporating confidence measures enables interpretations and conclusions to be appropriately conservative. Pie charts at nodes to represent ancestral state estimates of categorical traits inherently provide some sense of confidence as the likelihood of each state is readily visible in the pies. Similarly, ancestral state estimates for binary traits can be plotted over the tree as branch colours reflecting probabilities [103], but methods for continuous traits are typically less straightforward and available. Nevertheless, they do exist for some methods for continuous traits, such as the use of shading around the point estimates of ancestral states [103] or the use of coloured bars illustrating 95% confidence intervals at nodes of ancestral state estimates [54].

## 10. Conclusions

Phylogenetic comparative methods are broadly applicable but rarely used by evolutionary toxinologists. Whenever a study uses an interspecific dataset, accounting for phylogeny (or making use of the information contained within phylogenies) is vital, and studies should be designed to be adequately powered to make strong inferences. More generally, comparative biology provides one route to extend the (often) descriptive nature of toxinological studies towards a solid hypothesis-testing approach by exploiting the natural variation we see in toxic organisms around the World. This review has attempted to provide an introduction to the range of questions in evolutionary toxinology that phylogenetic comparative methods can help to address, with the aim of familiarising researchers with what is possible. Nevertheless, like all methodological approaches, the range of methods discussed here have their own set of considerations, caveats, and limitations, explaining why comparative biology has become a discipline of evolutionary biology in itself. Hence, in order to fully exploit the benefits of this approach to toxinology, the importance of either seeking training in comparative methods or collaborating with comparative biologists should not be underestimated.

## Figures and Tables

**Figure 1 toxins-10-00518-f001:**
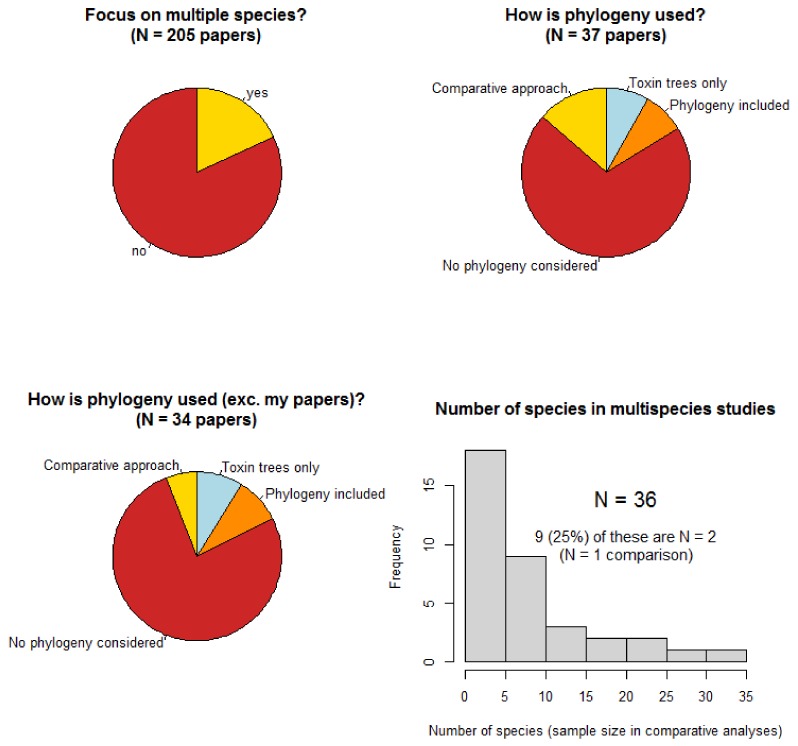
Summary of research articles published in the Animal Venoms section of *Toxins*. The top-left pie chart displays the proportion of papers focusing on multiple species, while the top-right pie chart shows the proportion of those which use phylogenies in different ways. The bottom-left pie chart is the same as the previous one but with my own publications removed to show that the lack of comparative biology is magnified in the field more generally. The histogram in the bottom right shows the number of species used in multispecies papers, highlighting the generally insufficient sample sizes to conduct reliable inference. Note that one outlier publication of mine is excluded here which analysed data from 19,161 tetrapod species [11].

**Figure 2 toxins-10-00518-f002:**
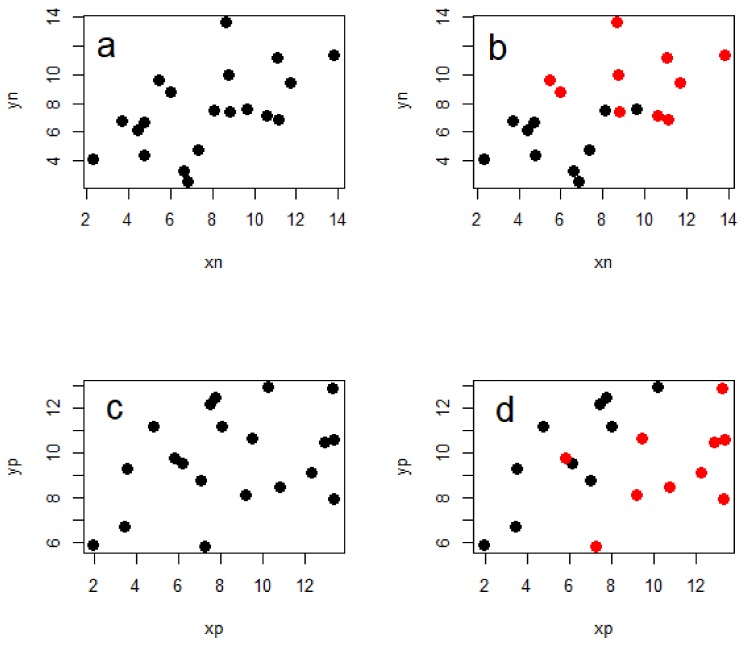
Simplified representation of the need to account for phylogeny. These four plots each show the relationship between two continuous variables and each have 20 data points. The data in each plot come from two different clades—red and black—each with 10 species. Note that (**a**) and (**b**) are identical except that (**b**) has data from the two clades shown in different colours, and the same applies to (**c**) and (**d**). In plot (**a**) you would falsely conclude that there is a positive relationship between the two traits, but in (**b**) we can see that this is simply a result of the data coming from two different clades with different trait values—no relationship is evident in either of the clades. In contrast, there seems to be no relationship in (**c**), however plot d clearly shows a strong positive relationship in each of the two clades which is only likely to be recovered when accounting for phylogeny in the analysis.

**Figure 3 toxins-10-00518-f003:**
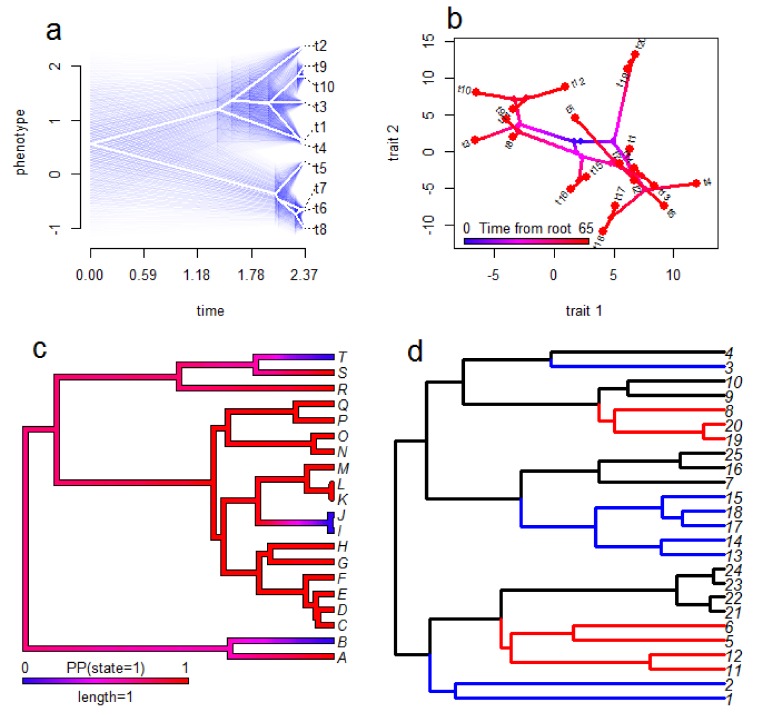
Examples of visualisation methods for comparative biology. These examples are based on simulated data and are not comprehensive, but illustrate some possibilities for some of the methods discussed in this review. The ‘traitgram’ in panel (**a**) shows the evolution of the value of a single continuous trait (y-axis) over time (x-axis) based on ancestral state reconstruction with 95% confidence intervals shown as blue transparencies to indicate uncertainty. In panel (**b**) a ‘phylomorphospace’ plot is shown to illustrate the relationship between two continuous traits in the context of their phylogenetic history. The colour scheme of the phylogeny shows the time in which blue represents the time at the clade’s origin and red is the present day—from this we can clearly see the expansion of trait diversity over time. Panel (**c**) shows an ancestral state reconstruction for a two-state categorical trait with the colour scheme representing the probability of being in state 1 (as opposed to state 0). We can see that the ancestral state is likely to be 1 whereas three independent origins of state 0 have occurred over the phylogeny. Panel (**d**) displays the results of an analysis of convergent evolution using SURFACE (see Section 6), with different colours representing different regimes of trait evolution (not trait values) and shared colours reflecting convergent evolution of the same regime. In this example the blue regime has evolved independently three times and the red regime has evolved twice independently.

**Table 1 toxins-10-00518-t001:** Examples of introductory reviews for further information on each of the groups of methods covered in each section of the current review, and of useful R packages for each group of methods. Sections are given an abbreviated title and their number for clarity. Numbers in the ‘introductory review(s)’ column refer to the numbers in the reference section. References for R packages are given only on their first mention, and packages mention under ‘general comparative biology’ are useful for most of the other sections but are not repeated for each for brevity.

Section	Introductory Reviews	R Packages
General comparative biology	[10,13,15,16,17,18,19,20]	ape [21]; phytools [22]
Accounting for phylogeny (2)	[23,24]	caper [25]; MCMCglmm [26]; phylolm [27]
Estimating ancestral states (3)	[28,29]	corHMM [30]; diversitree [31]
Trait evolution models (4)	[17,19,20]	corHMM; diversitree; geiger [32]
Evolutionary pathways (5)	[33,34,35]	corHMM
Convergent evolution (6)	[36,37,38]	convevol [39]; surface [40]; windex [41]
Diversification dynamics (7)	[31,42,43,44]	BAMMtools [45]; diversitree; geiger

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
