# Peer review of "Phylogenetic Comparative Methods can Provide Important Insights into the Evolution of Toxic Weaponry"

_toxins, 2018, doi:10.3390/toxins10120518_

Round 1

Reviewer 1 Report

To whom it may concern:

The manuscript under consideration attempts to introduce phylogenetic comparative methods and an explicit evolutionary approach to the study of venom toxins. The effort is largely successful, but I do have a couple of suggestions for improvement.

1.      The focus on only interspecific relationships is too limiting. Comparative biology and PCMs are appropriate at other phylogenetic/taxonomic scales. For example, evolutionary comparisons between populations is a very interesting avenue for those studying venoms, and PCMs can be used to strengthen these studies.

2.      In general, I think the paper would benefit from more detailed examples. Despite the goal of introducing these complex methods to an audience, more concrete examples (including perhaps the authors own work) would help to solidify the utility of these approaches. Additional figures would help to accomplish this too. The current manuscript feels more like a lite literature review than a truly persuasive argument.

Despite these concerns, this manuscript is quite strong and well researched. I am excited to see the final version and think it will be a boon to the study of venoms. Please see additional minor issues below:

Line 228: add a comma after “tree” and replace “down” with “done”.

Line 329: the end of this sentence needs rewriting

Line 370: “involved”

Line 376: “… doesn’t need subduing” or “… doesn’t need to be subdued”? Also, remove the contraction.

Line 377-379: this sentence needs citations.

Line 403: add a comma after the citation

Line 429: “… but see Arbuckle et al. [37] for…”

Line 434: add a comma after “single trait”

Line 441-443: This sentence needs rewriting.

Line 459-460: subject verb agreement issues between “group” and be verbs

Line 505: sentence fragment

Author Response

I thank all reviewers for their encouraging and thoughtful comments on our manuscript. I hope that we have satisfactorily addressed their concerns. Where text from the manuscript is quoted, I have highlighted changes in bold unless the entire section is new.

Reviewer comments in red. Our response in blue.

Reviewer 1 comments

"The manuscript under consideration attempts to introduce phylogenetic comparative methods and an explicit evolutionary approach to the study of venom toxins. The effort is largely successful, but I do have a couple of suggestions for improvement.”

“Despite these concerns, this manuscript is quite strong and well researched. I am excited to see the final version and think it will be a boon to the study of venoms”

I thank the reviewer kindly for these positive comments, I hope I have addressed the suggestions below adequately.

"1. The focus on only interspecific relationships is too limiting. Comparative biology and PCMs are appropriate at other phylogenetic/taxonomic scales. For example, evolutionary comparisons between populations is a very interesting avenue for those studying venoms, and PCMs can be used to strengthen these studies."

The reviewer is of course correct that comparative methods can be applied to subspecific (e.g. phylogeographic) analyses as well as interspecific datasets. The focus on the latter is due to 1) interspecific datasets being what these methods are intended to address and by far the most common area where phylogenetic information should be included but often isn’t; 2) there are many exceptions but typically populations will have higher levels of gene flow than that which occurs across a reasonably size clade of species, so the assumptions of many comparative methods (e.g. increasing dissimilarity with increasing phylogenetic distance) are likely to be met in more cases for interspecific data; 3) interspecific phylogenies are more frequently available than intraspecific ones, and while I would encourage construction of new phylogenetic trees for studies where possible it is commonplace to make use of previously published trees, so the generality of focussing on interspecific datasets is greater. That said, the initial point still stands and I agree with it, so I have included text to highlight this in the introduction as follows:-

“Note that data from, and intraspecific phylogenies of, different populations are equally amenable to analysis with comparative methods. Hence, the focus on the more common case of interspecific data in this article should not be taken as indicating that intraspecific phylogenetic data are unimportant as the same comments apply where there is important interpopulation structure.”

"2. In general, I think the paper would benefit from more detailed examples. Despite the goal of introducing these complex methods to an audience, more concrete examples (including perhaps the authors own work) would help to solidify the utility of these approaches. Additional figures would help to accomplish this too. The current manuscript feels more like a lite literature review than a truly persuasive argument."

I attempted to provide brief examples at the end of each section to tie the concepts back to the ‘real world’, partly for brevity and partly to avoid the very broad suite of concepts and methods discussed in each section being reduced to one or two examples that might give a restricted impression of their applicability. One issue of putting together a persuasive argument more in the style of a narrative rather than a literature review is that few studies have compared results from analyses with and without phylogeny because either the latter isn’t possible in many cases or (for regression-style methods) it is explicitly advised against presenting both approaches (see Freckleton, R.P. The seven deadly sins of comparative analysis. J Evol Biol, 2009, 22, 1367-1375). Hence, the more ‘review’ style is intended to give toxinologists an overview of what kind of thing is available and directions to further sources of information to follow up on for questions they might want to pursue. Section 1 attempts to state what the problem is and why it’s important to address, but the other sections (and the table) essentially aim to provide a roadmap for entry into the vast literature - an in-depth practical guide to what is a large field of evolutionary biology is beyond the scope of this article (as stated at the end of the introduction).

Nevertheless, I have expanded slightly on the examples given in the various sections with a focus on highlighting how the use of comparative methods enhances research above standard approaches. This will hopefully reinforce and persuade the reader that comparative methods aren’t only essential for interspecific (and some intraspecific) studies, but also useful and capable of providing new insights that are impossible with other approaches.

"Line 228: add a comma after “tree” and replace “down” with “done”"

Done.

"Line 329: the end of this sentence needs rewriting"

I’ve replaced ‘modelled’ with ‘modelling’ and added the word ‘analysis’, which I think addresses the issue: “a pattern in which older venomous lineages (whose toxins been evolving for longer) show slower rates of toxin evolution than younger venomous lineages has been recovered in a broad-scale analysis [58], though without explicit modelling of a slowdown in trait evolution”.

"Line 370: “involved”"

Done.

"Line 376: “… doesn’t need subduing” or “… doesn’t need to be subdued”? Also, remove the contraction."

Changed to “does not need to be subdued”.

"Line 377-379: this sentence needs citations"

Thanks for pointing this out, added 3 relevant citations now.

"Line 403: add a comma after the citation"

Done.

"Line 429: “… but see Arbuckle et al. [37] for…”"

Done.

"Line 434: add a comma after “single trait”"

Done.

"Line 441-443: This sentence needs rewriting."

Reworded as follows: “SURFACE allows the quantification of the frequency of convergence via the number of convergent shifts, the number of different convergent regimes, and the proportion of regimes that are convergent”.

"Line 459-460: subject verb agreement issues between “group” and be verbs"

I’ve split this sentence into three sentences which (I think) are grammatically better :-

“The Wheatsheaf index combines two aspects of trait evolution. Firstly, it considers convergence to be stronger when members of the focal group are more similar to each other. Secondly, convergence is considered to be stronger when the focal group are more dissimilar to the non-focal group, as this implies a stronger selective pull across the adaptive landscape”.

"Line 505: sentence fragment"

Fixed by adding the word ‘exist’ to the sentence: “Many comparative methods exist for investigating diversification dynamics…”.

Reviewer 2 Report

In the interest of scientific openness and with the objective of helping the author to communicate his work in the most effective way, I have waived my right to anonymity.

Major Observations

It appears that the author is well qualified to address this subject, or so it seems to one who is unqualified to assess the methods he is discussing from a technical standpoint.  The topics he has discussed (main headings in the manuscript) would indeed be of interest to many researchers.  However, while the author has targeted his manuscript at “biological toxinologists and toxicologists” who are not yet using the statistical methods he describes here, most of the manuscript is written to address the cognoscenti instead.  He has missed the mark completely.  The paragraph starting on Line 185 provides a good example.  The paper needs to be heavily rewritten for its stated audience, because in its present state, it communicates very little to that audience.

Section 7 is more accessible to the target audience and it contains a lot of interesting ideas; however, it could also be fleshed out with more examples and written so that it communicates to a broader audience. An interesting case is related in Lines 501-502, but the author drops it without developing it.  Similar things could be done in this and the subsequent section.

Section 8 is perhaps the most interesting and useful part of the paper, but the author fails to be as didactic as he could, which is his ostensible purpose.  We are always looking for intuitive ways to relate the import of patterns in complex datasets.  If he would provide examples of such graphics and carefully explain their strengths and weaknesses, this would be very useful!

The quality of the writing also needs considerable improvement.  It is colloquial, excessively wordy, contains awkward expressions, and also contains numerous grammatical and punctuation errors.  The author would do well to seek professional editing assistance to improve his technical writing skills.

Minor Details and Examples

Line 2.  It is true that we are steadily advancing in our understanding of venom composition, but pharmacologically we know almost nothing about what most of these do.  We have even less understanding of their synergistic interactions.

Lines 5-6.  “The literature on chemical weaponry of organisms is vast and provides a rich understanding of the composition and mechanisms of the toxins and other components involved.”  You speak of toxins and other components.  Some venom proteins are non-toxic.  How are you using this term.  What are the other components?

Line 8.  Be more specific. What do you mean by “an ecological view”?  Several studies have considered the influence of prey type.

Line 31.  explicit focus on… ecological aspects of venoms  One major obstacle to doing this is that so little information exists on prey taken by venomous snakes.

Lines 36-37.  Be more explicit.  What sorts of ecological interpretation?

Lines 49-53  However, many phylogenies have been based on only one or two loci, such as mitochondrial sequences, and that seems too easy and too flimsy.  In part, the problem is that we cannot easily delineate what constitutes a species.

Lines 94-97  These revised statistics can probably be dropped.

Line 116.  It is unclear what sorts of variables you mean.

Line 130.  This is not a personal letter.  Address the issue, not the reader.  Also, throughout the manuscript you are in first person singular, first person plural, second person, and third person.  Pick a voice and stay there.

Page 158-160.  It might be a good idea not to be so condescending toward your readers.

Line 470.  How does what relate?

Lines 496-497.  Explain this more fully.

Lines 525-526.  How will people who do not know these techniques accomplish this?

Lines 528-529.  Don’t introduce it.  Just address it.

Lines 532-533.  After observing it, then what do you suggest?  We can observe 2D gels without statistical methods.  Teach us something new.

Lines 556-557.  This is an important point, but it has little impact.  Show us some creative examples and explain their interpretation in a well-written figure legend.

Lines 561-562.  Don’t just cite the reference.  Show us an example so as to illustrate your point.

Author Response

I thank all reviewers for their encouraging and thoughtful comments on our manuscript. I hope that we have satisfactorily addressed their concerns. Where text from the manuscript is quoted, I have highlighted changes in bold unless the entire section is new.

Reviewer comments in red. Our response in blue.

“In the interest of scientific openness and with the objective of helping the author to communicate his work in the most effective way, I have waived my right to anonymity.”

I appreciate this approach, though I can’t actually view the reviewer’s name so anonymity has been preserved. That said, although I echo the reviewer’s views on open vs closed peer review, it should make little difference to my responses here as I hope to address the reviewers concerns satisfactorily.

"It appears that the author is well qualified to address this subject, or so it seems to one who is unqualified to assess the methods he is discussing from a technical standpoint.  The topics he has discussed (main headings in the manuscript) would indeed be of interest to many researchers.  However, while the author has targeted his manuscript at “biological toxinologists and toxicologists” who are not yet using the statistical methods he describes here, most of the manuscript is written to address the cognoscenti instead.  He has missed the mark completely.  The paragraph starting on Line 185 provides a good example.  The paper needs to be heavily rewritten for its stated audience, because in its present state, it communicates very little to that audience."

Based on the example given I respectfully disagree here (and note that reviewer 1 came to the opposite conclusion to reviewer 2 here). The review cannot provide a detailed introduction to statistical analysis generally, and doesn’t attempt to do this – it is an introduction to comparative biology not a primer on standard statistical data analysis. Indeed, as the intended audience are scientists it assumes familiarity with basic statistical concepts and aims to introduce toxinologists to a different class of methods for data analysis beyond standard statistical methods. The example given of the paragraph starting on line 185 deals with standard concepts such as model residuals, independence of observations as an assumption of statistical analysis, and highlights phylogenetic equivalents of basic statistical models such as logistic regressions. Other than familiarity with these concepts the paragraph does not assume any specialist knowledge. It does use the term ‘GLM’ (general linear model) but the paragraph immediately preceding this one explains what this is and puts it into the context of other commonly used tests such as t-tests, ANOVA, and regressions. Given that these core concepts should be familiar to anyone engaged in analysing data, falling under the standard material covered in undergraduate science modules in data analysis and any textbook for biostatistics, and the non-phylogenetic tests mentioned are commonly used in papers published in Toxins, I don’t think the level is aimed at an inappropriately high level. I could direct the reader to introductory textbooks for statistics in biology if the reviewer still feels that these concepts are beyond the average reader of Toxins, but this would seem to me an overly condescending addition.

"Section 7 is more accessible to the target audience and it contains a lot of interesting ideas; however, it could also be fleshed out with more examples and written so that it communicates to a broader audience. An interesting case is related in Lines 501-502, but the author drops it without developing it.  Similar things could be done in this and the subsequent section."

I have provided a little more detail on this finding (and on other examples, see above response to reviewer 1 for additional relevant comments). However, I believe that extended discussion of the results of previous studies that I have used as examples seems a little tangential to the focus of the paper, which is the kind of questions that can be asked rather than what previous answers have been. The following additional text has been added to the examples highlighted by the reviewer: “The reasons for the increased extinction rate of poisonous amphibians over evolutionary time is still unknown, but persists into the present day as increased probability of having a threatened conservation status [73]. These results open up a new avenue for research to understand not just why poisonous amphibians suffer higher extinction risk, but why this differs from other toxic tetrapod groups. This results would not be clear nor would its further exploration be possible without an explicitly phylogenetic approach.”

In terms of more examples, I have included at least one example in each section but bear in mind there are a relatively small number of (toxinological) examples to choose from as this is an underused approach in toxinology, which is of course the motivation for the review.

"Section 8 is perhaps the most interesting and useful part of the paper, but the author fails to be as didactic as he could, which is his ostensible purpose.  We are always looking for intuitive ways to relate the import of patterns in complex datasets.  If he would provide examples of such graphics and carefully explain their strengths and weaknesses, this would be very useful!"

I agree that this is a promising area for future work in evolutionary toxinology, but as explained in that section, I can’t provide fully worked and illustrated examples as the method doesn’t yet exist. The proper development of new methods would typically comprise a paper in its own right, or at least a substantial methods section in a paper describing results from its use. The idea of suggesting one possible approach in a relatively cursory way is simply to highlight the potential for such methods and indicate the extendible nature of (comparative) method development, which can be tailored to the unique properties of toxinology in cases where current approaches do not help to answer the question.

"The quality of the writing also needs considerable improvement.  It is colloquial, excessively wordy, contains awkward expressions, and also contains numerous grammatical and punctuation errors.  The author would do well to seek professional editing assistance to improve his technical writing skills."

In addition to addressing specific issues highlighted by reviewer 1 (see above) I have gone over the manuscript again and corrected many parts of the text. I have kept the writing style as originally written as although ‘colloquial’ I do not feel it is excessively informal and much literature has emphasised the benefits of a less formal scientific writing style to comprehension by a broad audience (e.g. Biber and Gray, 2010, ‘Challenging stereotypes about academic writing: complexity, elaboration, explicitness’, Journal of English for Academic Purposes 9:2-20; and recommendations in Matthews et al., 1996, Successful Scientific Writing, Cambridge University Press). I thank the reviewer for drawing my attention to several errors that I was able to correct as this has helped to improve the manuscript substantially.

"Line 2. It is true that we are steadily advancing in our understanding of venom composition, but pharmacologically we know almost nothing about what most of these do.  We have even less understanding of their synergistic interactions."

I agree, but saying toxinology has revealed much about the composition and mechanisms of toxins is not the same as saying ‘we know everything about all toxins’ or that there is nothing left to learn.

"Lines 5-6.  “The literature on chemical weaponry of organisms is vast and provides a rich understanding of the composition and mechanisms of the toxins and other components involved.”  You speak of toxins and other components.  Some venom proteins are non-toxic.  How are you using this term.  What are the other components?"

You are correct, some venom proteins (and other molecules) are non-toxic, so when I refer to ‘toxins and other components’ I mean the components of chemical weaponry that are not toxins (i.e. are non-toxic). As this isn’t a review on biochemistry but on a methodological approach (which can be applied equally well to toxins or non-toxic components depending on the interest of the researcher) I have not attempted to go into depth on the particular components of animal venoms and poisons. That seems distinctly beyond the scope and outside the aims of the review and I am not quite sure what the reviewer wants me to change here.

"Line 8.  Be more specific. What do you mean by “an ecological view”?  Several studies have considered the influence of prey type."

I don’t feel it is the place of an abstract to go into depth like this, and the specific interpretation of ‘ecological’ here will vary with the researcher’s interests and question (but can clearly apply to any aspect of the animal’s ecology). However, that sentence is related to molecular evolution studies, some of which may ‘consider’ the influence of prey type but these are rare (coevolution of diet and venom is usually addressed separately from studies of the molecular evolution of toxins) and often fall into the patterns highlighted in the introduction (e.g. “with a post hoc ecological or evolutionary interpretation added to discussions” rather than explicitly addressed as part of the formal analyses in the paper).

"Line 31.  explicit focus on… ecological aspects of venoms  One major obstacle to doing this is that so little information exists on prey taken by venomous snakes."

I agree entirely, but I am not sure what the reviewer wants me to change as the point of the sentence is not dependent on the reasons for the stated trend.

"Lines 36-37.  Be more explicit.  What sorts of ecological interpretation?"

I have added in an example as follows, which I hope does not detract from the flow of the paragraph since: “For instance, some aspect of the venom of a group of snakes could be described and the results may show that one species is an outlier in some way. The researcher may be tempted to look for unique attributes of that species, and noting that it is the only bird-eater in the group interpret the venom differences in terms of adaptations to feeding on birds.”

"Lines 49-53 However, many phylogenies have been based on only one or two loci, such as mitochondrial sequences, and that seems too easy and too flimsy.  In part, the problem is that we cannot easily delineate what constitutes a species."

This is very true, but a separate point from that being made. The fact that phylogenies should be of good quality does not detract from the fact that molecular phylogenies of individual toxins are not species phylogenies (particularly as they are likely to be under strong selection) or that species phylogenies are necessary to fully understand the evolution of venom systems. Furthermore, even though most comparative methods assume the phylogeny is known (though uncertainty can be taken into account in various ways for different methods) it is likely that imperfect phylogenetic information is better than no information at all. Ignoring phylogeny definitely doesn’t reflect the evolutionary history of the system (as highlighted in Felsenstein’s classic 1985 paper in Am Nat on ‘Phylogenies and the comparative method), incorporating it improves our inference at least to some degree (unless the whole phylogeny is so bad to provide absolutely no information). There is nothing in the text that suggests we should not strive to use the best phylogenies possible. Clearly, as for all forms of data, we should aim to use the best quality data we can.

"Lines 94-97  These revised statistics can probably be dropped."

I agree they could be dropped, but given I am writing this review to highlight a wider issue in the field (and my work which focusses on this subject could consequently bias the picture) I think they help emphasise the lack of comparative biology in the current toxinology literature and contribute little extra space. Hence, unless there are strong counter-reasons to remove them I think they add to the point being made.

"Line 116.  It is unclear what sorts of variables you mean"

I am not referring to any particular variables – the “variables of interest” will depend on the interest of the researcher who is conducting the study and the question they are trying to address. The aim of this review is to provide a general introduction to an entire class of methods that can be applied to a wide range of questions so a certain degree of general language is both necessary and, I believe, important to emphasise the generality of the point itself. Bear in mind the paper is not intended as a review of all the ways comparative biology has been used (although it uses examples where possible to highlight these), but as a review of the potential ways it can be used, so discussion of variables is a general concept that can be applied to any given context, not just limited to one particular variable.

"Line 130.  This is not a personal letter.  Address the issue, not the reader.  Also, throughout the manuscript you are in first person singular, first person plural, second person, and third person.  Pick a voice and stay there."

I disagree that addressing the reader is inappropriate for scientific prose (see the references on scientific writing above for instance), but I have changed this here and have ensured that, where relevant, all text is written in first person active voice (as the majority was and this voice is widely regarded as the most appropriate for scientific writing, including by several eminent researchers such as Robert May, former president of the Royal Society).

"Page 158-160.  It might be a good idea not to be so condescending toward your readers."

The text in question is as follows: “Nevertheless, comparative biology is a specialism in itself and I encourage toxinologists to either receive some degree of training in comparative biology or consult a comparative biologist before running such analyses (and preferably at the earlier stage of planning the study).” My apologies, but I still do not see this as being condescending at all. To the contrary, I think it is a reasonable piece of advice and important to point out that unlike standard statistical analyses which scientists will be familiar with, comparative biology is a distinct methodological specialism that is difficult to jump into with little understanding (though many do). Given this advice is not standard practice amongst toxinologists or many other biologists but is no different than my own practice of collaborating with biochemists when I’m interested in chemical processes, I fail to see how it is condescending.

"Line 470.  How does what relate?"

In common with other section headings, this one is framed as a general question of the form that can be addressed by the methods discussed in the section. Once again, this review is not about specific findings for specific traits, but about a general analytical approach to answer a broad array of questions. With this in mind, the generality of wording in subject headings and many other places seems appropriate.

"Lines 496-497.  Explain this more fully"

At the risk of derailing the point being made I have added one more sentence to concisely give a little more detail as follows: “These far-reaching effects are predicted to result from mimicry enabling harmless species to occupy more ‘dangerous’ (higher predation risk) niches and lifestyles which may enhance diversification”.

"Lines 525-526.  How will people who do not know these techniques accomplish this?"

As I explained above in the text the reviewer earlier considered ‘condescending’, these techniques are best integrated into toxinology via one of two routes: 1) by toxinologists seeking training in comparative biology, or 2) seeking collaborations with comparative biologists to solve problems and answer questions in an interdisciplinary way. I’m not sure what the reviewer wants me to change here.

"Lines 528-529.  Don’t introduce it.  Just address it."

This comment seems to refer to this text: “As way of explanation of the ‘new workflow’ approach I will provide a suggestion that could be pursued in more detail in future work”. Omitting this sentence would remove my caveat that I am not intending to fully develop this new method here (which would be beyond the scope of the review) but rather illustrate the type of approach to new method development that could be adopted more generally. Hence I believe it would be misleading to exclude this ‘introductory’ sentence as it provides necessary context for the suggestion that follows.

"Lines 532-533.  After observing it, then what do you suggest?  We can observe 2D gels without statistical methods.  Teach us something new."

The very next sentence states “In the field of animal coloration methods have been developed to quantify the overall similarity of patterns (e.g. the distance transform method [94]), and these could likely be applied equally well to the patterns on a gel”. The answer to the reviewer’s comment is therefore ‘we can quantify the pattern similarities using techniques such as the distance transform method’. The paragraph then goes on to provide very general suggestions of how these quantified ‘venom profile similarities’ can be further analysed (deliberately kept general as, again, the details will depend on the question being asked which will depend on the researcher’s specific question in a given study). Again, I’m not sure what the reviewer want me to change here.

"Lines 556-557.  This is an important point, but it has little impact.  Show us some creative examples and explain their interpretation in a well-written figure legend."

Thanks for this suggestions, I’ve included a new figure (Fig. 3) with examples of visualisation options and referred to it at the end of the relevant sentence. A detailed interpretation of each of the examples would result in an excessively long legend (it’s already quite lengthy) but I have tried to give enough detail to illustrate the type of information that can be conveyed by each plot.

"Lines 561-562.  Don’t just cite the reference.  Show us an example so as to illustrate your point."

I’m not clear what kind of example the reviewer would like me to include here – the text being referred to simply states that phylogenetic regression lines may not appear to fit the data well on a plot as the plotted raw data are not themselves adjusted for phylogeny. As far as I can see an ‘example’ would just be a plot with a (apparently) poorly fitting regression line added, which would seem to add no more than a description of the same idea and so be a largely uninformative plot. If the reviewer has particular ideas of what kind of example they have in mind I’d be glad to consider them.

Finally I would like to thank the reviewers and the editor once more for their time and comments. I firmly believe that their efforts have improved the manuscript.

Round 2

Reviewer 2 Report

I have no further suggestions to make.